# Recovering Immunogenic *Orthohantavirus puumalaense* N Protein from Pellets of Recombinant *Escherichia coli*

**DOI:** 10.3390/vaccines13070744

**Published:** 2025-07-10

**Authors:** Natalya Andreeva, Ekaterina Martynova, Polina Elboeva, Milana Mansurova, Ilnur Salafutdinov, Aleksandr Aimaletdinov, Rafil Khairullin, Diksha Sharma, Manoj Baranwal, Sara Chandy, Dilbar Dalimova, Alisher Abdullaev, Mirakbar Yakubov, Albert Rizvanov, Svetlana Khaiboullina, Yuriy Davidyuk, Emmanuel Kabwe

**Affiliations:** 1Institute of Fundamental Medicine and Biology, Kazan (Volga Region) Federal University, 420008 Kazan, Russia; natasha_andreeva_2000@mail.ru (N.A.); ignietferro.venivedivici@gmail.com (E.M.); polinaelboeva@mail.ru (P.E.); chirkova.milana@yandex.ru (M.M.); sal.ilnur@gmail.com (I.S.); aimaletdinowam@gmail.com (A.A.); rafil.khairullin@gmail.com (R.K.); rizvanov@gmail.com (A.R.); sv.khaiboullina@gmail.com (S.K.); davi.djuk@mail.ru (Y.D.); 2Department of Medical Biology and Genetics, Kazan State Medical University, 420012 Kazan, Russia; 3Department of Biotechnology, Thapar Institute of Engineering and Technology, Patiala 147001, India; sdiksha946@gmail.com (D.S.); baranwal.manoj@gmail.com (M.B.); 4The Childs’ Trust Medical Research Foundation (CTMRF) Kanchi Kamakoti Childs Trust Hospital (KKCTH), Chennai 600034, India; sarachandy@yahoo.co.in; 5Center for Advanced Technologies, Tashkent 100174, Uzbekistan; d.dalimova@cat-science.uz (D.D.); abdullaev_alisher@yahoo.com (A.A.); mirakbardan@yahoo.com (M.Y.)

**Keywords:** N protein, hemorrhagic fever with renal syndrome, *Orthohantavirus puumalaense*, vaccine, immunogenicity, immunization

## Abstract

(1) Background: Hemorrhagic fever with renal syndrome (HFRS) remains a prevalent zoonosis in Eurasia. *Orthohantavirus puumalaense* (PUUV), carried by bank voles (*Myodes glareolus*), is the principal zoonotic pathogen of HFRS in this region. Despite ongoing efforts to develop effective drugs and vaccines against PUUV, this challenge remains. (2) Aim: In this study, we aimed to express a large quantity of the PUUV recombinant N (rN) protein using *E. coli.* We also sought to develop a protocol for extracting the rN protein from pellets, solubilizing, and refolding it to restore its native form. This protocol is crucial for producing a large quantity of rN protein to develop vaccines and diagnostic tools for HFRS. (3) Methods; PUUV S segment open reading frame (ORF) coding for N protein was synthesized and cloned into the plasmid vector pET-28 (A+). The ORF was transformed, expressed and induced in BL21(DE3) pLysS *E. coli* strain. Subsequently, rN protein was purified using immobilized metal affinity and ion chromatography. Immune reactivity of rN protein was tested by employing in house and commercial VektoHanta-IgG kit ELISA methods (both in vitro and in vivo). (4) Results: The best conditions for scaling up the expression of the PUUV rN protein were an incubation temperature of 20 °C during a 20 h incubation period, followed by induction with 0.5 mM IPTG. The most significant protein yield was achieved when the pellets were incubated in denaturing buffer with 8M urea. The highest yield of refolded proteins was attained using non-denaturing buffer (50 mM Tris-HCl) supplemented with arginine. A final 50 μL of PUUV rN protein solution with a concentration of 7 mg/mL was recovered from 1 L of culture. The rN protein elicited an antibody response in vivo and reacted with serum taken from patients with HFRS by ELISA in vitro. (5) Conclusion: Therefore, the orthohantavirus N protein’s ability to elicit immune response in vivo suggests that it can be used to develop vaccines against PUUV after conducting in vitro and in vivo studies to ascertain neutralising antibodies.

## 1. Introduction

*Orthohantaviruses* are zoonotic pathogens circulating in small rodents [1,2,3,4,5]. Hemorrhagic fever with renal syndrome (HFRS) is a disease caused by several members of *orthohantaviruses* [1,4,6]. In the Russian Federation, *Orthohantavirus puumalaense* (PUUV) is a primary pathogen for HFRS [7,8], which accounts for a significant proportion of zoonotic disease cases [9]. Despite ongoing efforts, developing an effective vaccine against PUUV remains challenging [10].

The genome of *Orthohantaviruses* has three segments of negative-sense, single-stranded RNA: small (S), medium (M), and large (L) [9]. These RNA segments encode the nucleocapsid (N) protein (433 amino acids (*aa*)), the glycoprotein precursor (Gn and Gc, 1138 *aa*), and the viral RNA-dependent RNA polymerase (RdRp, 2155 *aa*) [11]. The N protein encapsidates the genomic RNA (gRNA), forming a ribonucleocapsid that protects the viral nucleic acid from degradation [12]. The ribonucleocapsid also plays an instrumental role in initiating the virion assembly and prevents intra-strand base pairing within the gRNA template [13]. This pivotal role of the N protein in the encapsidation and assembly of the gRNA underscores its significance in the context of immunogenicity [14].

The N protein is the first and most abundant protein synthesized after infection [13,15]. This protein is also a potent immunogen, as anti-N protein antibodies are detected soon after the onset of the disease symptoms [15,16]. Therefore, the N protein is the primary diagnostic antigen used in multiple laboratory tests to detect anti-orthohantavirus antibodies [15,17]. The expression of virus N protein in *Escherichia coli* (*E. coli*) encounters several challenges, such as protein toxicity and the formation of insoluble aggregates (pellets) upon overexpression [18]. Multiple steps of the purification should be followed to achieve a substantial protein yield. These steps consist of: (1) codon optimization, (2) selection of a vector (plasmid), (3) compatibility between the *E. coli* strain and the vector, (4) the expression conditions, (5) the optimization of centrifugation protocol, (6) the selection of suitable solubilizing agents, and (7) the utilization of appropriate refolding buffers. Ultimately, selecting suitable chromatographic columns to purify the protein from the aggregated protein masses remains a critical component of the process [19].

This study aims to express a large quantity of the PUUV recombinant N (rN) protein using *E. coli.* We also sought to develop a protocol for extracting the rN protein from pellets, solubilizing, and refolding it to restore a native form. This protocol is crucial in producing a large quantity of rN protein to use in the development of vaccines against PUUV.

## 2. Materials and Methods

### 2.1. Bacteria Strain and Plasmid for the Expression of rN Protein

The PUUV S segment open reading frame (ORF) coding for N protein was obtained from a bank vole (MG2458 strain, GenBank accession # PP112638) captured in the Republic of Tatarstan, Russia. The ORF was synthesized and cloned into the plasmid vector pET-28 (A+) by Evrogen (Evrogen, Moscow, Russia), which allows the expression of recombinant protein with 6 × His-tag at the N-terminal end. The BL21(DE3) pLysS *E. coli* strain was used for transformation with the recombinant pET-28 (A+) plasmid. The transformed *E. coli* was cultured in a medium containing Luria Bertani (LB), following the standard expression protocol. The BL21 (DE3) pLysS *E. coli* strain was selected for its compatibility with the T7 promoter in the expression vector.

### 2.2. Codon Optimization of PUUV S ORF

Codon optimization was completed using the GenSmart codon optimization tool, an openly accessible online resource [20]. This step was implemented to improve rN expression in *E. coli*. Moreover, the optimization was essential to prevent rho-independent transcription termination, bacterial ribosome binding issues, and the restoration of enzyme cleavage sites.

### 2.3. In Silico Prediction of PUUV rN Protein Properties

Protein solubility and toxicity were analyzed using the SoluProt [21] and ToxinPred tool [22], respectively. SoluProt uses a gradient boosting machine technique with the Target Track database as a training set. The aggregation-prone regions in the PUUV protein were predicted using the AMYPRED-FRL tool [23].

### 2.4. Expression and Solubilization of rN Protein

The expression of the rN protein was completed in three steps. First, transformed *E. coli* was cultured overnight in 50 mL of LB medium at 37 °C and 180 rotations per minute (RPM), supplemented with 50 µg/mL of kanamycin as a selection marker. Then, the culture was scaled up to 1 L (L) of LB medium and was grown at 37 °C to an OD600 = 0.6. Finally, protein expression was induced with 0.5 mM IPTG for 20 h at 20 °C.

Then, the culture was rapidly cooled on ice and centrifuged at 4000× *g* for 15 min at 4 °C to harvest the cells. The samples were sonicated at 4 °C (12 cycles at 70% amplitude with the space of 30 s pause) with Buffer 1 (Table 1) and cell lysates were clarified by centrifugation 24,400× *g* for 30 min. The supernatant and pellet were used for the sodium dodecyl-sulfate polyacrylamide gel electrophoresis (SDS-PAGE) analysis. The rN protein was found only in the pellet.

The pellet was washed three times: twice with Buffer 2 and once with Buffer 3 (Table 1). The buffers contain a low concentration of urea as a chaotropic agent and Triton X-100 to remove the membrane surrounding the insoluble protein [24]. Following each wash, the pellet was harvested by centrifugation at 2900× *g* for 30 min. Afterwards, the pellet was resuspended in sterile deionized water and subjected to two consecutive centrifugations at 2900× *g*.

The pellet was divided into three portions (Portions 1, 2, and 3) and each was used to identify the best solubilizing agent. Portion 1 was solubilized in Buffer 4 (substantial denaturation), Portion 2 in Buffer 5 (mild denaturation), and Portion 3 in Buffer 6 (non-denaturing solution containing dimethyl sulfoxide (DMSO)). The chemical composition of buffers is summarized in Table 1. The pellet was incubated for 24 h at 22 °C under stirring conditions, with minimal agitation at 90 RPM. Samples were collected at 2 h intervals to determine the concentration of the released rN protein from each buffer spectrophotometrically at 280 nm, using the DeNovix DS11+ spectrophotometer (DeNovix Inc., Wilmington, DE, USA). At the end of incubation, the rN protein was pelleted at 24,400× *g* for 30 min, with subsequent analysis of the supernatant and the pellet using SDS-PAGE.

### 2.5. Dialysis of PUUV rN

Buffers 7, 8, and 9 were used to restore the rN protein’s native state (Table 1). Solubilized proteins were refolded by the removal of solubilizing agents and dialysis of the proteins with Buffers 7, 8, and 9. Adding arginine and glycerol could enhance the stability of the protein [25]. Dialysis was completed at 22 °C under stirring of the outer phase with minimal agitation. To optimize the refolding process of the recombinant protein, Buffers 7, 8, and 9 were added at 4 h intervals. The dialysis continued for 24 h, when rN protein was pelleted by centrifugation at 24,400× *g* for 30 min. Both, the supernatant and pellet were analyzed using SDS-PAGE. The protein concentration was determined spectrophotometrically at 280 nm using DeNovix DS11+ (DeNovix Inc., Wilmington (DE), USA).

### 2.6. Analysis of rN Protein Antigenicity

The flat bottom 96-well plate was coated with rN protein (50 ng/100 μL PBS) and placed at 4 °C for 18 h. Subsequently, wells were washed three times with the 300 μL buffer containing 0.01MPBS and 0.5% Tween 20 (PBS-T). Plate was blocked with 300 μL of PBS-T supplemented with 5% fat-free milk (PBS-T-milk) and left for 1 h 30 min at 37 °C. Then, wells were washed 3 times and HFRS serum dilution of 1:200 in 100 μL PBS-T-milk, respectively, was added followed by incubation for 90 min at 37 °C. Next, plate was washed 3 times and anti-human IgG-HRP conjugated antibodies (1:10,000; American Qualex, Orange, CA, USA) were added to the wells for 90 min at 37 °C. After being washed (3 times; PBS-T), wells were incubated with 3,3′,5,5′ Tetramethylbenzidine substrate (Chema Medica, Moscow, Russia) for 10 min. The reaction was stopped by adding the stop solution (2 M sulfuric acid) and analyzed using a microplate reader, Tecan 200 (Tecan Group Ltd., Männedorf, Switzerland), at an optical density of 450 nm with reference at OD650 nm.

### 2.7. Immunization of Mice with PUUV rN Protein

Mice were immunized two times at one-week intervals (at day 0 and at day 7) (Figure 1). Serum samples were collected on day 0, day 7, and day 14 post-immunization. The rN protein was mixed in a ratio of 1:1 with a complete Freunds adjuvant. Briefly, four mice were immunized subcutaneously with 50 µg of rN protein, while another four mice received 100 µg of protein. One week later, serum samples were collected, and a second injection of rN protein with 50 µg and 100 µg was completed, both diluted in PBS. After one week, serum samples were collected. Also, two mice were not immunized and were used as a control (control). The serum collected before the first immunization was used as an additional control. All collected serum samples were stored at −80 °C until the analysis.

### 2.8. Animal Ethics Statement

Institutional Review Board Statement: The Kazan Federal University (KFU) Animal Care and Use Committee approved all animal-use procedures (protocol #23, dated 30 June 2020).

Mice (10 weeks old) were ordered from the breeding facility of the Kurchatov Institute (Rappolovo, Russia). All animals received food and water ad libitum and were contained in a room with 12 h light/12 h dark cycle with an ambient temperature of 22 °C at KFU animal center. Small mammals were euthanized by exposing them to CO_2_ using the gradual filling method at a displacement rate of 30% to 70% of chamber volume/min.

### 2.9. Human Subject Ethics Statement

The Ethics Committees of the Kazan State Medical Academy (KSMA) approved this study (protocol 4/09, dated 26 September 2019, and KFU by Article 20 of Federal Law “Protection of Health Rights of Citizens of the Russian Federation” N323-FZ, dated 21 November 2011). Informed consent was diligently obtained from each participating patient and control subject as outlined in the respective protocols.

### 2.10. Serum Samples Collection

Blood samples were collected from 27 patients diagnosed with HFRS during an outbreak in 2019, the Republic of Tatarstan. Patients were hospitalized in the Agafonov Republican Clinical Hospital for Infectious Diseases, Republic of Tatarstan. The mean age of the HFRS patients was 39.6 ± 13.6 years. All HFRS serum samples were also tested using the VektoHanta-IgG kit (Vekto Best, Ufa, Russia) to confirm the detection of anti-orthohantavirus antibodies. Additionally, blood samples from 27 controls (38 ± 16.3 years) were obtained. All controls had no prior history of HFRS and tested negative for anti-orthohantavirus antibodies. Serum samples from both groups were stored at −80 °C.

### 2.11. Enzyme Linked-Immunosorbent Assay (ELISA) Analysis of Serum Samples

IgG to PUUV rN protein was analyzed using the commercial kit (Vektor-Best, Ufa, Russia) with a few modifications. A serum sample (100 µL; 1:100 dilution) was added to wells pre-coated with orthohantavirus antigens (1 h at 37 °C), washed (5 times, 400 µL PBS-T), and incubated with goat anti-mouse IgG-HRP (1:10,000; American Qualex technologies, San Clemente, CA, USA) secondary antibodies for 30 min at 37° C. The reaction was visualized by adding 3,3′,5,5′ Tetramethylbenzidine (Vektor-Best, Ufa, Russia) for 15 min, followed by a stop solution. Results were analyzed using a Tecan 200 plate reader at OD450.

### 2.12. Statistical Analysis

All experiments were carried out in triplicate and are presented as the mean ± standard deviations. The data were processed using the GraphPad Prism 10. The distribution of the data was assessed using Shapiro–Wilk test. The data was considered statistically significant among the groups when *p*-values were less than 0.05, as assessed using the one-way ANOVA and adjustment for multiple comparisons with Benjamini–Hochberg method.

## 3. Results

### 3.1. Codon Optimization of the PUUV S Segment, Cloning of the ORF and Selection of E. coli Strain

The codon adaptation index (CAI) value for the S segment sequence was 0.72 after optimization, slightly below the recommended range of 0.78 to 1.0 [20]. The GC content was 70% following optimization, which is within 30% and 75% typical for scaling up the rN protein expression in *E. coli*. The optimized PUUV S segment ORF was cloned into the pET-28 (A+) vector and confirmed by PCR and sequencing using the Sanger method. Subsequently, the cloned vector was transformed into the BL21 (DE3) pLysS strain of *E. coli.*

### 3.2. Expression of the rN Protein and Isolation of Pellets

To screen the optimal culture conditions, transformed *E. coli* was maintained under various culture conditions such as temperature, incubation time, and different isopropyl β-d-1-thiogalactopyranoside (IPTG) concentrations (0.02–0.7 mM) (Table 2). *E. coli* was harvested, pelleted, and used to analyze the rN protein on the SDS-PAGE (Figure 2). The obtained protein has a molecular mass of approximately 50 kDa, which is expected for PUUV N protein (Figure 2B).

The pure protein was obtained using chromatography, which is influenced by various factors such as pellet formation that requires solubilization, and subsequent protein refolding [18,19]. For instance, the IPTG concentration used to induce *E. coli* [26]. In this study, we noted that inducing *E. coli* with a high concentration of IPTG (more than 0.7 mM) led to overexpression of the protein. Similarly, the temperature adjustment was critical; incubating the cells at 37 °C was ideal for protein production but significantly reduced the protein yield from the pellet protein.

Thus, the optimal conditions for the maximum expression of the PUUV rN protein were determined: incubation temperature of 20 °C, a 20 h incubation period, and induction with 0.5 mM IPTG. Then, harvesting *E coli* by centrifugation at 4000× *g* for 30 min and sonication. The SDS-PAGE analysis revealed a higher amount of the PUUV rN protein in the pellets compared to the supernatant fraction (Figure 2B). Subsequently, the pellets were rinsed in sterile deionized water and centrifuged at 3000× *g* for 30 min. The resulting pellets were used to purify the PUUV rN protein. Also, the pellets were washed twice using washing Buffers 2 and 3 to enhance purity. The washing buffers used and their contents are summarised in Table 1. The washing steps helped separate pellets from aggregates, as other researchers have also demonstrated [18,24,27].

### 3.3. Solubilization of PUUV rN Protein from Pellets

The most substantial protein release was observed when the pellets were incubated in denaturation Buffer 4 (Figure 3, Buffer 4). The mild denaturing of Buffer 5 (Figure 3 Buffer 5) released less protein than the Buffer 4. The yield of rN protein was similar between mild denaturing Buffer 5 and non-denaturing Buffer 6 after 24 h of incubation (Figure 3, Buffer 6).

The quantity of released protein using denaturation Buffer 4 has increased during the first 8 h of incubation. Subsequently, the protein release rate decreased, and the level remained relatively stable after 12 h (Figure 3, Buffer 4). Our study determined that a higher urea concentration was sufficient to denature the PUUV rN protein and release it from pellets produced in *E. coli* (Figure 3, Buffer 4).

### 3.4. Refolding of the Solubilized PUUV rN Protein (Dialysis Method)

The rN protein was refolded using dialysis Buffers 7, 8, and 9. The composition of the refolding/dialysis buffers is summarized in Table 1. Notably, the highest yield of refolded proteins was attained using the non-denaturing Buffer 9 (Table 1) supplemented with arginine (Figure 4). This data suggests that adding arginine to the dialysis Buffer inhibits protein aggregate formation.

No substantial difference in the recovery of the rN protein was demonstrated when dialysis Buffer 7 and Buffer 8 were used (Table 1). However, adding glycerol to dialysis Buffer 8 hindered protein refolding, resulting in reduced recovery. The protein obtained after the dialysis had a molecular mass of about 50 kDa, expected for PUUV rN protein. The rN protein was subsequently purified using immobilized metal affinity chromatography (IMAC) (Bio-works Technologies AB, Uppsala, Sweden) and ion chromatography using WorkBeads^TM^ 40/100 SEC resin (Bio-works Technologies AB, Uppsala, Sweden) following the manufacturer’s instruction [28]. After chromatographic purification, purified protein samples were concentrated using Amicon ultrafiltration cells with a 30 kDa cutoff. Employing the optimized approach for protein purification, we recovered 50 ul PUUV rN protein solution with a concentration of 7 mg/mL from 1 L of *E. coli* LB culture.

### 3.5. Immune Reactivity of rN Protein with HFRS Serum

The PUUV rN protein immune reactivity was tested using HFRS serum collected during the 2019 outbreak in the Republic of Tatarstan. All HFRS serum samples reacted with the PUUV rN protein in ELISA (Figure 5 black bar chart. The seropositivity of HFRS serum samples was also confirmed with a commercially available VektoHanta-IgG kit (IgG kit) (Figure 5 grey bar chart). All control samples were anti-orthohantavirus antibody negative (Figure 5).

Next, we sought to determine the immune property of the rN protein. Mice were immunized with PUUV rN protein. Serum collected one and two weeks post-immunization was used to detect PUUV rN protein-specific IgG. Anti-PUUV rN protein IgG was not detected in pre-immunization serum samples (Figure 6A,B). In contrast, anti-PUUV rN protein IgG was present in the post-immunization serum. Notably, serum from mice immunized with 50 µg of rN protein had high IgG antibody OD compared to 100 µg after the first injection (Figure 6A). In contrast, mice immunized with 100 µg of rN protein had more antibodies after the second injection (Figure 6A). These data suggest that the rN protein can elicit a humoral immune response and, therefore, can be used to develop vaccines against PUUV.

## 4. Discussion

The insoluble aggregates (pellets) present a significant challenge when viral proteins are expressed in *E. coli*. These aggregates are formed when protein is overexpressed resulting in an unstable form [29]. The recovery and generation of the recombinant protein is a multifaceted and multistep process [30]. In our study, multiple steps and methods have been carefully optimized to augment the protein retrieval from the pellets and subsequently facilitate its refolding into its active conformation.

The expression of functional proteins outside their original context is often multifaceted. To mitigate codon bias, a codon optimization of the S segment was completed [20]. The CAI index value obtained in this study was slightly lower than recommended [20]. However, according to the “Index Definition of GenRCA Rare Codon Analysis Tool” (GenScript, Piscataway, NJ USA), this value is still within the range which could achieve the optimal protein expression [20]. Following codon optimization, the S segment ORF was cloned pET-28 (A+) vector. The pET series of plasmids are commonly used for the expression of recombinant proteins in *E. coli* [31]. These plasmid series have genetic modules controlling transcription and translation, required for optimal protein expression in *E. coli* [31]. Also, the selection of the BL21 (DE3) pLysS strain aligned with its compatibility with the pET series of plasmids [32].

The screening of the best conditions for scaling up the expression of the PUUV rN protein were: incubation temperature of 20 °C during a 20 h incubation period, followed by an induction with 0.5 mM IPTG. The most significant protein yield was when the pellets were incubated in denaturation Buffer 4 (Figure 3, Buffer 4). These conditions were used to express the N protein in *E. coli*. Previous studies reported that different conditions for the recombinant protein expression in *E. coli* can affect pellet formation [18,31,32]. Adding a glucose during pre-incubation was shown to have limited impact on the protein yield, as did the supplementation with ethanol during induction [18,19]. Oganesyan et al. found that heat shock increased the yield of soluble protein expressed in *E. coli*, whilst the insoluble protein yield was low [33]. Heat shock notably increased protein production, as reported by Lipnicanova et al. [18]. In our study, the heat shock did not facilitate the protein production; instead, it resulted in the ablation of the protein expression. Moreover, lowering the temperature to 15 °C decreased protein expression and pellet quantity.

Pellets were obtained at a selected condition and solubilised using higher concentrations of chaotropic reagents, such as urea and guanidine hydrochloride [19,26,34]. To investigate the impact of various solubilizing agents on PUUV rN protein, we used strong denaturing (Buffer 4 in Table 1), mild denaturing (Buffer 5, Table 1), and non-denaturing (Buffer 6, Table 1) buffers. The most substantial protein release was observed when pellets were incubated in strong denaturation Buffer 4 (Figure 3, Buffer 4). Our findings are consistent with the results of a previous study by Lipnicanova et al. in which they reported a high protein yield using similar solubilization conditions [18]. The quantity of protein released using strong denaturing Buffer 4 increased during the first 8 h of incubation. At the later time points, the protein release rate decreased, and the level remained relatively stable after 12 h (Figure 3, Buffer 4). The notable yield of proteins extracted using Buffer 4 can be attributed to the chaotropic nature of urea, which disrupts hydrogen bonding in recombinant proteins [19]. Furthermore, incorporating dithiothreitol (DTT) (5–100 mM) and EDTA into the denaturing buffers is known to disrupt disulphide bonds in the protein and prevent metal-catalyzed air oxidation of cysteines, respectively [19]. Singh et al. reported enhanced protein solubilization after adding lower concentrations of propanol and urea [35]. Our study determined that a higher urea concentration was sufficient to denature the PUUV rN protein and release it from pellets produced in *E. coli* (Figure 3, Buffer 4). The release of the rN protein from pellets, as illustrated in Figure 4, surpassed the yield reported when alternative expression systems were used [36,37].

Next, the released protein was refolded to restore its functional form. Refolding is achieved by removing chaotropic agents such as urea and other compounds used during the solubilization phase [38,39]. Interestingly, the highest yield of refolded proteins was attained using the non-denaturing Buffer 9 (Table 1) supplemented with arginine (Figure 4). There was limited difference in the recovery of the rN protein when dialysis Buffer 7 and Buffer 8 were used (Table 1). However, adding glycerol to dialysis Buffer 8 hindered protein refolding, resulting in reduced recovery. The protein obtained after the dialysis had a molecular mass of about 50 kDa, expected for PUUV rN protein. The recovery PUUV rN protein is higher than that purified from similar orthohantavirus- and non-orthohantavirus-related studies [18,40,41]. While many studies have demonstrated the expression and purification of orthohantavirus recombinant protein in *E. coli* [42,43,44], this study presents an optimized method for the production, solubilization and purification of the immunogenic PUUV rN protein. Also, this method produced a yield which was higher than those previously reported.

The purified rN protein was tested for immune properties in vitro and in vivo for potential use to develop vaccines against PUUV. The HFRS serum samples reacted with the PUUV rN protein in vitro and anti-PUUV rN protein IgG was found in the post-immunization serum in vivo. Our data confirm that the orthohantavirusrN protein expressed and purified in this study maintains its immunogenic properties. The elicitation of an antibody response in mice supports previous reports [1,45,46]. Additionally, the reactivity of rN protein with HFRS serum is in line with the results found by Piet Maes et al., demonstrating that the truncated rN protein specifically reacts with HFRS serum samples [47]. Among the viral proteins, N protein has immunodominant epitopes and robust immunogenic property, which trigger a humoral immune response [15,16]. The antigenicity of N protein is conserved compared to that of envelope glycoproteins [16]. Moreover, the N protein was found to induce protective immunity against PUUV in bank voles [48]. Also, the N protein in combination with Gn/Gc elicited a strong humoral immune response [49,50] and could induce neutralizing antibody against the virus [43]. Additionally, the N protein stimulates the T cell immune response which can be cross-reacting [51,52,53]. Further, the N protein of other othohantaviruses, DOBV and SEOV, was shown to induce antibodies in mice [54,55]. However, the efficacy of these recombinant proteins’ protection against orthohantaviruses remains to be determined [56]. In our study, we first purified the rN protein which then was used to study the immunogenicity. This approach differs from that previously used where only short amino acids of N protein of PUUV virus were used; in this study a whole N protein was used for immunization. Our data demonstrated that rN protein remains immunogenic, suggesting that it could be used for vaccine development [15,16]. In the last decades, the widely used vaccine for the prevention of orthohantavirus infections is Hantavax based on inactivated virus, which only protects against HNTV and SEOV, the orthohantaviruses endemic in Asia [1]. This vaccine could induce long-term immune response with antibodies detected 33 months post-immunization [1]. Currently, there is no vaccine against PUUV in Russia.

## 5. Conclusions

Collectively, our findings present a comprehensive method for harvesting a substantial quantity of recombinant PUUV N protein expressed in *E. coli*. The applied technique could serve as a foundation for expressing the N protein of other orthohantaviruses. HFRS serum demonstrated the immune properties and immunogenicity of the PUUV N protein. Additionally, immunizing mice confirmed its immunogenicity. Successful purification of the N protein and eliciting of immune responses in mice suggest that the PUUV rN protein could be used to generate subunit vaccines against PUUV after ascertaining its neutralizing properties. Also, the purification method employed in our study could be used for scaling up production of the N protein. However, the efficacy of these recombinant proteins’ protection against PUUV infection remains to be determined.

## Figures and Tables

**Figure 1 vaccines-13-00744-f001:**
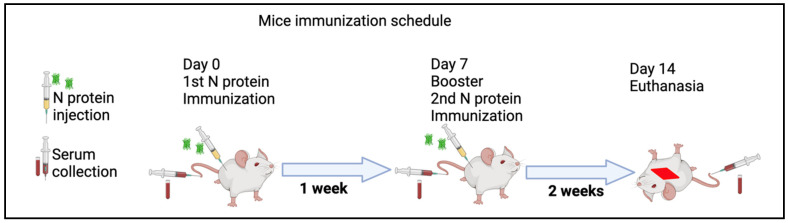
Immunogenicity assessment of recombinant N protein in vivo schematic experiments. Serum samples were collected from mice. Then, mice were inoculated subcutaneously with N protein (50 µg and 100 µg in Freunds adjuvant) at day 0. After one week (at day 7) post immunization serum, samples were collected and mice were again given a booster injection with the same dose of N protein mixed with PBS. Two weeks later (at day 14), serum samples were collected and animals were euthanized according to the ethical international principles. All collected serum samples were kept at −80 °C until analysis.

**Figure 2 vaccines-13-00744-f002:**
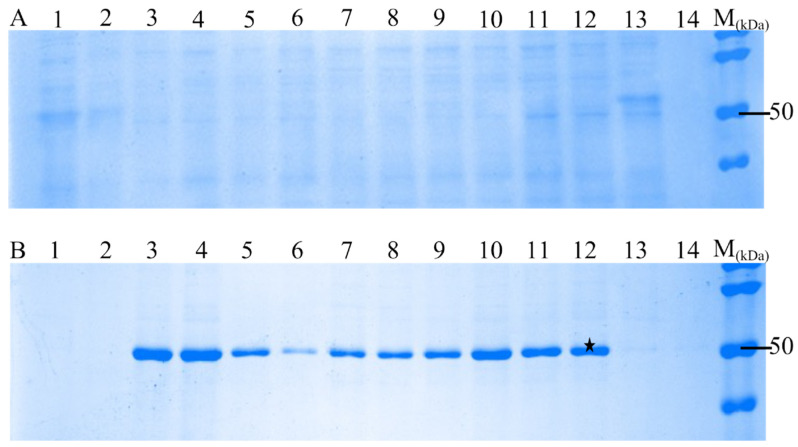
SDS-PAGE analyses of recombinant N protein expressions in *E. coli* (BL21 (DES) pLysS) strain using different culture conditions. Supernatant and pellet obtained after centrifugation were analyzed by SDS-PAGE using 12% gel. Two fractions obtained are (**A**)—supernatant analyzed by SDS-PAGE; (**B**)—pellet analysed by SDS-PAGE and showing the recombinant N protein (approximately 50 kDa). The best expression condition marked with a black star was selected for large expression of rN protein.

**Figure 3 vaccines-13-00744-f003:**
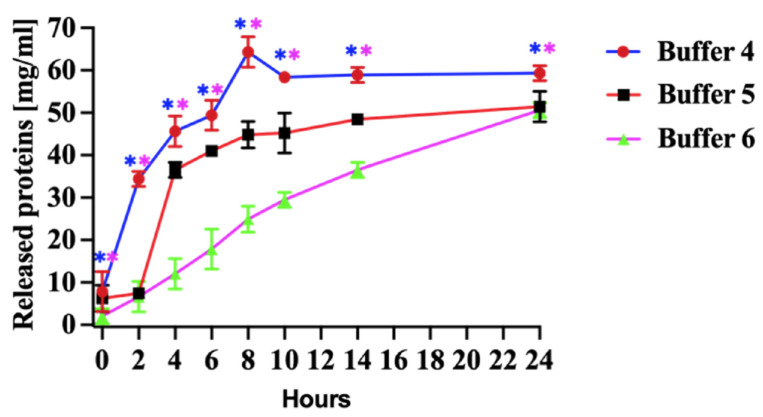
Released protein from pellets. Buffer 4, effect of denaturing (8 M urea), Buffer 5, mild denaturing (2 M urea and 6 M propanol) and Buffer 6, non-denaturing (5% (*v*/*v*) DMSO) solubilizing agents in Tris-HCl buffer (pH 8.0) on released proteins for 24 h. All experiments were carried out in triplicates. The released N protein concentration was measured at 280. The results are presented as the mean and standard deviations. Blue asterisks represent statistical significance between Buffer 4 and 6, and pink is between Buffer 4 and 5, *p* < 0.05.

**Figure 4 vaccines-13-00744-f004:**
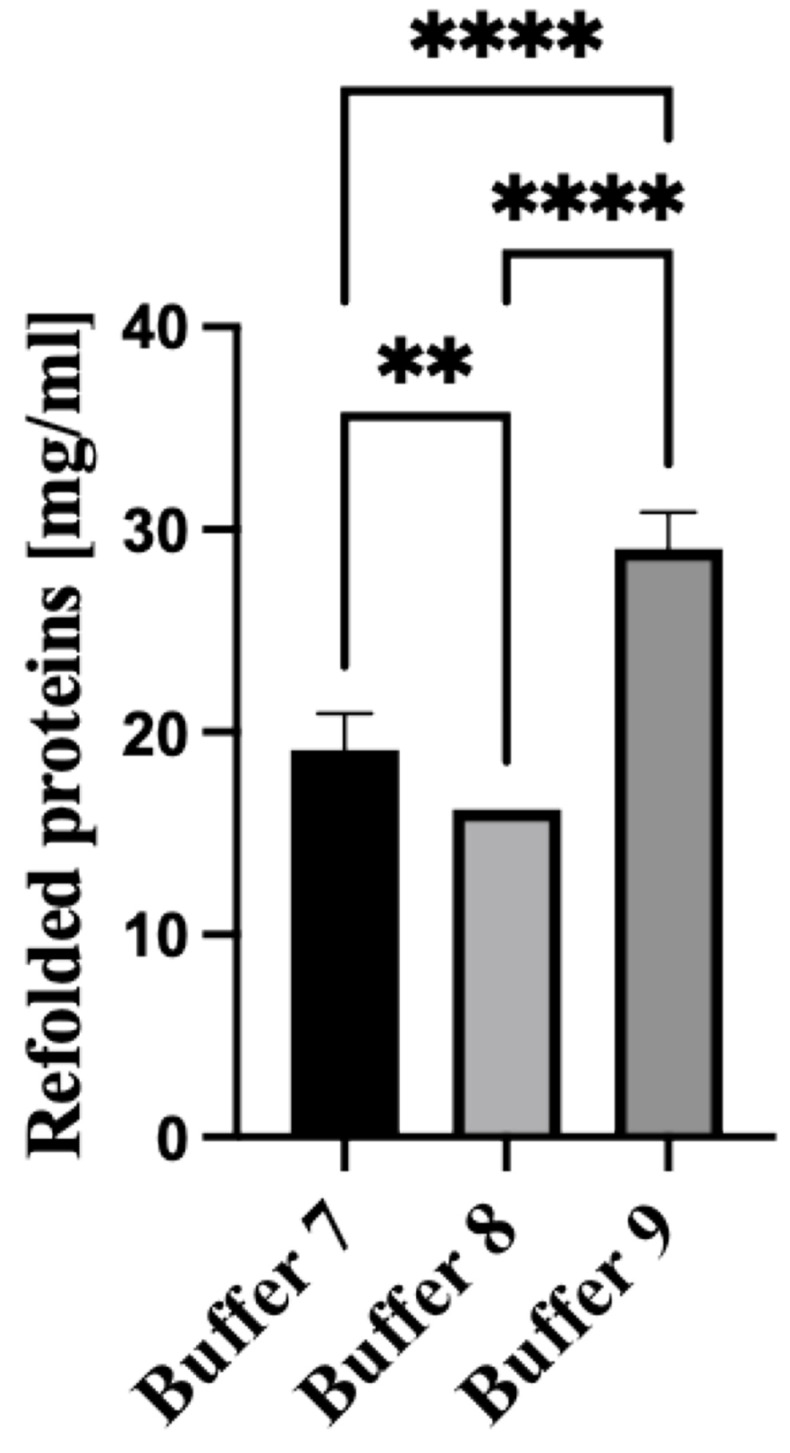
Recovery of denatured PUUV N protein using various dialysis buffers. Buffer 7 and 8 contain 150 mM NaCl and 5 mM EDTA, and in addition, Buffer 8 was supplemented with 0.3 M glycerol, whilst Buffer 9 had only 0.5 M arginine. All dialysis agents were dissolved in 50 mMTris-HCl buffer at pH 8.0. All experiments were carried out in triplicates. The released N protein concentration was determined spectrophotometrically at 280 nm using DeNovix DS11. The results are presented as the mean and standard deviations. For all statistical analyses, one-way ANOVAs with multiple comparisons were performed using log-transformed data. All data were significant with ** *p* < 0.005 and **** *p* < 0.0001.

**Figure 5 vaccines-13-00744-f005:**
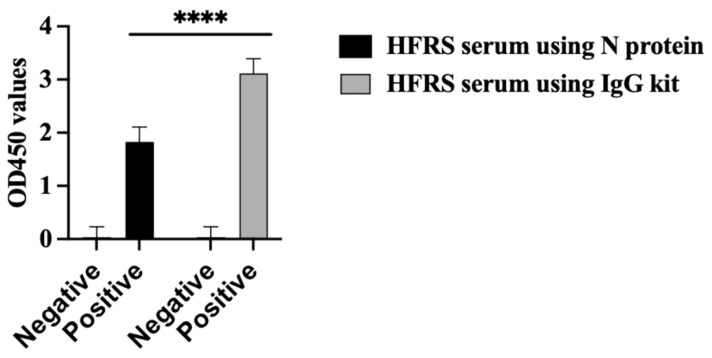
Immune reactivity of serum from HFRS patients. Shows the results of ELISA using N protein as an antigen coated on the plate and ELISA results using commercial kit (VektoHanta IgG kit). Twenty-seven (27) HFRS serum and twenty-seven control serum samples were tested for the presence of IgG antibodies. All experiments were carried out in triplicates. The results are presented as the mean and standard deviations of OD450 sample values. For all statistical analyses, one-way ANOVAs with multiple comparisons were performed using log-transformed data. All data were significant with **** *p* < 0.0001.

**Figure 6 vaccines-13-00744-f006:**
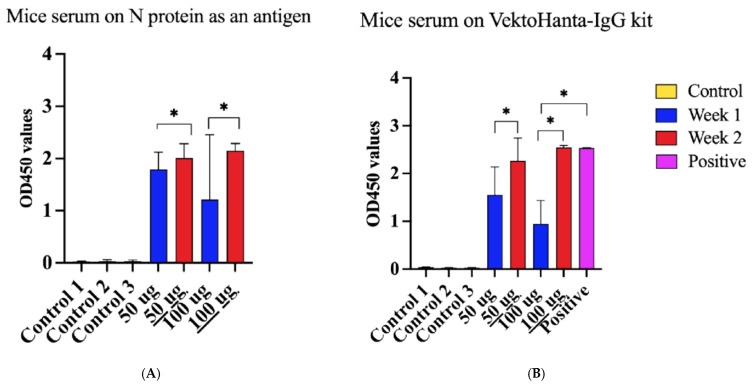
Reactivity of serum from HFRS patients and mice immunized with purified recombinant N Protein. (**A**) Immunoreactivity of serum from mice immunized with N protein using N protein as an antigen coated on the plate. Control 1 represents serum collected from 10 mice before immunization. Control 2 refers to serum collected from two (2) unimmunized mice after 1 week of the experiment, whilst Control 3 implies serum collected from 2 mice after two weeks of immunization. Additionally, 50 µg indicates the dose of N protein used to immunize four (4) mice and 100 µg is the dose of N protein used to immunize other four (4) mice; serum was taken after 1 week. 50 µg is the dose of N protein used to immunize four (4) mice and 100 µg indicates the dose of N protein used to immunize other four (4) mice; at the same time serum was taken after 2 weeks. (**B**) Reactivity of serum from immunized mice using VektoHanta-IgG kit, conducted following the manufacturer’s recommendations (with explanations same as Figure (**A**)). Positive control on Figure (**B**) represents the control provided in VektoHanta-IgG kit and was only used with the kit. For all statistical analyses, one-way ANOVAs with multiple comparisons were performed using log-transformed data. All data were significant with * *p* < 0.05.

**Table 1 vaccines-13-00744-t001:** Summary of buffers used for PUUV N protein purification.

Buffers Used for Initial Screening to Identify the Nature (Solubility) of the Expressed N Protein in *E. coli*					
Buffer No.	Content				Application
Sonication buffer					
	Content				
1	100 mM Tris-HCl/100 mMNacl	5 mMEDTA	1%Protease inhibitor cocktail	10 mM DTT/0.2 g/L Lysozyme/PH 8.1	

Solubilization and washing buffers (SoB and WB)					
	Content				
2	100 mM Tris-HCl	0.5%Triton X100			Washing
3	100 mM Tris-HCl	2 M Urea			Washing
4	100 mM Tris-HCl	8 M Urea	5 mM EDTA		Denaturation
5	100 mM Tris-HCl	2 M Urea	5 mM EDTA, 6 M propanol		Mild denaturation
6	100 mM Tris-HCl	5 mM EDTA	5% DMSO		Non denaturation

Buffers for refolding/dialysis					
	Content				
7	50 mM Tris-HCl	150 mM NaCl	5 mM EDTA		
8	50 mM Tris-HCl	150 mM NaCl	5 mM EDTA	0.3 M Glycerol	
9	50 mM Tris-HCl	0.5 M Arginine			

**Table 2 vaccines-13-00744-t002:** Culture tested conditions for the optimized expression of recombinant N protein.

	Temperature	IPTG (mM)	Incubation Time (Hours)	Medium (50 mL)
1	15 °C	0.02	20	LB
2	20 °C			
3	37 °C			
4	15 °C	0.3	24	
5	20 °C			
6	37 °C			
7	15 °C	0.5	15	
8	20 °C			
9	37 °C			
10	37 °C	0.7	20	
11	20 °C	0.5	20	LB + 0.5 M NaCl
12	20 °C	0.5	20	LB + glucose
13	40 °C Heat shock	0.5	30 min	LB
14			3	

## Data Availability

The original contributions presented in the study are included in the article, further inquiries can be directed to the corresponding author.

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
