# Peer review of "Recovering Immunogenic Orthohantavirus puumalaense N Protein from Pellets of Recombinant Escherichia coli"

_vaccines, 2025, doi:10.3390/vaccines13070744_

Round 1

Reviewer 1 Report

Comments and Suggestions for Authors

In this study, N protein of PUUV was expressed in E. coli. The inclusion body of N protein was dissolved in solubilization buffer and refolded in refolding/dialysis buffer. The antigenicity and immunogenicity of recovered N protein were tested. Below are some concerns and questions that should be clarified.

1)

Line 128, “Analysis of rN protein immunogenicity”

In this section, the reactivity of the rN protein with HFRS serum was tested. It is the analysis of protein’s antigenicity, not the immunogenicity. So, the title should be revised to “Analysis of rN protein antigenicity”.

2)

Line 148, “(50 μg PBS and 100 μg/PBS)”

This expression is not professional.

3)

Line 154, “Serum samples were collected from female mice and were inoculated subcutaneously with N protein”

Line 155, “After one-week post immunization at day 7”

Line 157, “Two weeks later at day 14”

English language needs revised.

4)

Line 211, “Following the optimization, the codon adaptation index (CAI) value for the S segment sequence was 0.58”

What was the codon adaptation index (CAI) value of N gene in E. coli before optimization. Why is CAI value so low after optimization? Was the GenSmart codon optimization tool used correctly? Usually, the CAI value would be greater than 0.8 after optimization.

5)

Figure 1: There are 14 lanes in Fig. 1A, while 13 lanes in Fig. 1B except the lane of protein Marker. In my understanding, after IPTG induction, the E. coli was harvested and sonicated. After centrifugation, both the inclusion body and the supernatant from each E. coli sample were analyzed by the SDS-PAGE. So, the number of lanes in Fig. 1A should be the same as that in Fig. 1B.

And each lane should be numbered in the new Fig. 1 of revised manuscript and the expression condition corresponding to each lane should be described in the Fig. 1’s caption or in the Table 3.

The number of lanes is much less than the number of conditions used in Table 3. So, It is very important to label each lane with description to tell the reader which conditions are displayed in Fig. 1.

Why do the protein Markers in Fig. 1A and Fig. 1B look different?

6) Line 237, “Total of 10 ul for each supernatant and pellet”

The pellet (inclusion body) is a solid. It is inappropriate to use “10 ul” for its measurement.

7) Line 246, “E. coli with a high concentration of IPTG (more than 0.7 mM) led to overexpression”

However, there is no condition with IPTG more than 0.7 mM in Table 3.

8) Line 247 and Line 249: “(Figure 3 marked with asteric 1).”

However, there is no asteric 1 in Figure 3.

9) Because both Figure 2 and Figure 3 display the distribution of N protein in the supernatant and inclusion body, there is no need to show two figures. All of the conditions to be shown to the readers should be displayed in one figure.

10) It was shown in Figure 3 that “higher amount of the PUUV rN protein in the inclusion body compared to the supernatant fraction” (Line 263). Why not purify N protein from the supernatant although the amount in supernatant is lower, which would be much easier than purification from the inclusion body. And the structure of N proteins in the supernatant is more likely to be correct than that refolded after denaturing and dialysis.

11) Line 285: “between mild denaturing Buffer 4”

Mild denaturing Buffer is Buffer 5.

12) Line 289: “B, mild denaturing (2 M urea and 6 M propanol) (Buffer 5)”

In Table 2, there is no 6 M propanol in Buffer 5.

13) Check the number of buffers in the caption of Fig. 5

14) Line 326: “Employing the optimized approach for protein purification, we recovered 30 ± 5 mg/ml of PUUV rN protein, which is higher than 2.5 mg/L of the N protein reported by Jonsson et al.”

And as shown in Fig. 7, the concentrations of refolded N protein in Buffer 7, Buffer 8 and Buffer 9 are all greater than 15 mg/ml.

Based on my limited experience, refolded protein tends to aggregate at such high concentration. Please check carefully whether the concentration is correct. In addition to measuring refolded N protein concentration at A280, SDS-PAGE should be employed for confirmation. Proteins of known concentration, such as BSA standard or protein Marker, should be run with the refolded N protein together in the same SDS-PAGE gel for estimating N protein concentration.

15) Line 382: “The rN protein was subsequently purified using nickel column affinity chromatography (Bio-works, Sweden) and ion chromatography techniques strictly following the standard procedures”

The purification method by nickel column affinity chromatography and ion chromatography should be described in the Materials and Methods section.

16) Line 348: “The results are presented as the mean and standard deviations of OD650 sample values.”

Is it OD650 or OD450?

17) Line 353: “Notably, serum from mice immunized with 50 μg of rN protein had high IgG antibody OD after the first injection compared to the second injection (Figure 7A).”

Is Figure 7A like this?

18) Why is the standard deviation of positive in Fig. 7A much larger than that the standard deviation of positive in Figure 6B?

Comments on the Quality of English Language

Minor editing of English language is required. Read the entire manuscript thoroughly for revision, especially the sentences I mentioned above.

Author Response

The authors would like to thank the reviewer for the insightful revision of our manuscript. Kindly find point by point the revision of the recommended points. All the suggestions have been incorporated into the main text.

  • Line 128, “Analysis of rN protein immunogenicity”

 In this section, the reactivity of the rN protein with HFRS serum was tested. It is the analysis of protein’s antigenicity, not the immunogenicity. So, the title should be revised to “Analysis of rN protein antigenicity”.

Agree: The authors would like to thank the reviewer for this observation, the title was corrected (Line 126).

Line 148, “(50 μg PBS and 100 μg/PBS)”

Agree: This particular line was corrected (Line 145).

 This expression is not professional.

 3)Line 154, “Serum samples were collected from female mice and were inoculated subcutaneously with N protein”

Agree: We apologise for the misleading sentences; this sentence was modified (Line 151).

Line 155, “After one-week post immunization at day 7”

Agree: The sentence was modified (Line 152).

Line 157, “Two weeks later at day 14”

Agree: the sentence was corrected (Line 154).

 English language needs revised. 

4) Line 211, “Following the optimization, the codon adaptation index (CAI) value for the S segment sequence was 0.58”

What was the codon adaptation index (CAI) value of N gene in E. coli before optimization. Why is CAI value so low after optimization? Was the GenSmart codon optimization tool used correctly? Usually, the CAI value would be greater than 0.8 after optimization. 

Agree: corrected. The correct figure (0.72) has been incorporated into the main text (Line 198).

5)Figure 1: There are 14 lanes in Fig. 1A, while 13 lanes in Fig. 1B except the lane of protein Marker. In my understanding, after IPTG induction, the E. coli was harvested and sonicated. After centrifugation, both the inclusion body and the supernatant from each E. coli sample were analyzed by the SDS-PAGE. So, the number of lanes in Fig. 1A should be the same as that in Fig. 1B.

Agree: the authors would like to thank the reviewer for this observation, the lanes were corrected (Figure 2).

And each lane should be numbered in the new Fig. 1 of revised manuscript and the expression condition corresponding to each lane should be described in the Fig. 1’s caption or in the Table 3.

The number of lanes is much less than the number of conditions used in Table 3. So, It is very important to label each lane with description to tell the reader which conditions are displayed in Fig. 1.

Agree: the lanes were labeled according to the table (Table 2).

 Why do the protein Markers in Fig. 1A and Fig. 1B look different?

Agree: the markers were corrected. Replaced by the right market in the experiment (Figure 2).

 6) Line 237, “Total of 10 ul for each supernatant and pellet”

 The pellet (inclusion body) is a solid. It is inappropriate to use “10 ul” for its measurement.

Agree: Corrected, the text was modified (Line 214).

 7) Line 246, “E. coli with a high concentration of IPTG (more than 0.7 mM) led to overexpression”

 However, there is no condition with IPTG more than 0.7 mM in Table 3.

Agree: The authors apologize for this mistake, the table was corrected (Table 2).

 8) Line 247 and Line 249: “(Figure 3 marked with asteric 1).”

 However, there is no asteric 1 in Figure 3.

Agree: The authors modified this part of the sentence (the sentence was deleted).

 9) Because both Figure 2 and Figure 3 display the distribution of N protein in the supernatant and inclusion body, there is no need to show two figures. All of the conditions to be shown to the readers should be displayed in one figure.

Agree: The authors have deleted figure 3, only figure 2 was left.

 10) It was shown in Figure 3 that “higher amount of the PUUV rN protein in the inclusion body compared to the supernatant fraction” (Line 263). Why not purify N protein from the supernatant although the amount in supernatant is lower, which would be much easier than purification from the inclusion body. And the structure of N proteins in the supernatant is more likely to be correct than that refolded after denaturing and dialysis.

Agree: the protein concentration in supernatant was very low and the attempt to purify it proved unfulfilling. Moreover, we further used this protein for immunization so we wanted higher concentration of the protein.

 11) Line 285: “between mild denaturing Buffer 4”

 Mild denaturing Buffer is Buffer 5.

Agree: The authors would like to thank the reviewer for this observation. The Buffer was corrected (Line 240).

 12) Line 289: “B, mild denaturing (2 M urea and 6 M propanol) (Buffer 5)”

 In Table 2, there is no 6 M propanol in Buffer 5.

Agree: 6 M propanol was added to the table (Table 1).

 13) Check the number of buffers in the caption of Fig. 5

Agree: The Buffers numbers were corrected (Figure 5).

 14) Line 326: “Employing the optimized approach for protein purification, we recovered 30 ± 5 mg/ml of PUUV rN protein, which is higher than 2.5 mg/L of the N protein reported by Jonsson et al.”

 And as shown in Fig. 7, the concentrations of refolded N protein in Buffer 7, Buffer 8 and Buffer 9 are all greater than 15 mg/ml.

 Based on my limited experience, refolded protein tends to aggregate at such high concentration. Please check carefully whether the concentration is correct. In addition to measuring refolded N protein concentration at A280, SDS-PAGE should be employed for confirmation. Proteins of known concentration, such as BSA standard or protein Marker, should be run with the refolded N protein together in the same SDS-PAGE gel for estimating N protein concentration.

Agree: The final total of recovered rN protein is 7mg/ml. The addition of arginine, time consideration, and other chaotropic agents helped to stabilize the protein from forming aggregates (Line 278).

15) Line 382: “The rN protein was subsequently purified using nickel column affinity chromatography (Bio-works, Sweden) and ion chromatography techniques strictly following the standard procedures”

 The purification method by nickel column affinity chromatography and ion chromatography should be described in the Materials and Methods section.

Agree: We agree with the reviewer’s observation. However, the authors would like to ask the reviewer, if possible, we can leave the links in the text to the methods because we followed strictly the standard well known procedures both for nickel column affinity chromatography and ion chromatography.

 16) Line 348: “The results are presented as the mean and standard deviations of OD650 sample values.”

 Is it OD650 or OD450?

Agree: OD450 (Line 290).

 17) Line 353: “Notably, serum from mice immunized with 50 μg of rN protein had high IgG antibody OD after the first injection compared to the second injection (Figure 7A).”

 Is Figure 7A like this?

Agree: The authors would like to thank the reviewer for this observation, the sentence was corrected (Line 298).

 18) Why is the standard deviation of positive in Fig. 7A much larger than that the standard deviation of positive in Figure 6B?

Agree: The authors would like to thank the reviewer for this observation. The most probable the standard deviation is high in fig7A is that the VektoHanta kit is optimized for identifying hantavirus IgG antibodies in human. In this study, we did not carry protocol optimization on serum mice. However, we will be developing the ELISA protocol where we will carry out the optimization.

Reviewer 2 Report

Comments and Suggestions for Authors

Orthohantaviruses are zoonotic pathogens that cause Hemorrhagic Fever with Renal Syndrome (HFRS) in small rodents, with the N protein serving as the primary diagnostic and potentially protective antigen. In this study, the authors aim to express large-scale recombinant N (rN) protein using E. coli and develop a protocol for extracting this protein from insoluble aggregates (pellets or inclusion bodies). They established optimal conditions for the culture and optimized procedures and buffers for refolding and recovering this protein to its native form. Finally, they demonstrated that the final protein product has similar efficacy as a diagnostic reagent in vitro and as an immunogen in mice, holding potential for developing vaccines and diagnostic tools for HFRS. This is a successful study and may be worthy of publication in Vaccine. However, I have several comments listed below:

1.     They claimed that this protocol allows for large-scale production of the rN protein, but there is no concrete evidence to confirm it. For example, how much yield of the final production of rN protein can they routinely get from 1L culture?

2.     In the Materials and Methods section, the tables for Plasmid (vector) pET-28 (a+), characteristics of the expressing host (E. coli), and specific proteases used for tag removal can be removed or simplified in the text.

3.     In the Results section:

    • “3.1. Codon optimization of the PUUV S segment”
    • “3.2. Cloning of the optimized S gene ORF and selection of E. coli strain”
    • “3.3. Expression of the rN protein and isolation of pellets”

These should be simplified or some content moved to the Methods section.

4.     In Table 3, please specify the scale of the experiment (e.g., 1L).

5.     In Figure 2, please label each lane.

6.     In Figure 4, please combine the three panels into one panel in curve format and provide statistical analyses to compare the results among these three different buffers.

7.     In Figures 5 and 6, please include statistical results.

8.     In Figure 7, please combine these two panels and provide a statistical comparison of these two immunogens.

Comments on the Quality of English Language

Acceptable

Author Response

The authors would like to thank the reviewer for the insightful revision of our manuscript. Kindly find point by point the revision of the recommended points. All the suggestions have been incorporated into the main text.

  1. They claimed that this protocol allows for large-scale production of the rN protein, but there is no concrete evidence to confirm it. For example, how much yield of the final production of rN protein can they routinely get from 1L culture?

Agree: The authors would like to thank the reviewer for this observation, for 1L culture we managed to purify 7000mg/L. The quantity was added to the text (Line 278).

  1. In the Materials and Methods section, the tables for Plasmid (vector) pET-28 (a+), characteristics of the expressing host (E. coli), and specific proteases used for tag removal can be removed or simplified in the text.

Agree: The table was removed.

  1. In the Results section:
    • “3.1. Codon optimization of the PUUV S segment”
    • “3.2. Cloning of the optimized S gene ORF and selection of E. coli strain”
    • “3.3. Expression of the rN protein and isolation of pellets”

These should be simplified or some content moved to the Methods section.

Agree: The authors would like to thank the reviewer for this observation. The sections were edited (Lines 196-203).

  1. In Table 3, please specify the scale of the experiment (e.g., 1L).

Agree: The table was corrected; 50 ml was added to the table. 50 ml induction final volume was used for screening of the conditions (Table 2).

  1. In Figure 2, please label each lane.

Agree: The figure was labeled (Figure 2).

  1. In Figure 4, please combine the three panels into one panel in curve format and provide statistical analyses to compare the results among these three different buffers.

Agree: Combined and the p value added to the legend of the figures. The analysis was statistically significant at the p value 0 ****P ≤ 0.0001 (Lines 249-250) (Figure 4).

  1. In Figures 5 and 6, please include statistical results.

Agree: The p value added to the legend of the figures. The analysis was statistically significant at the p value 0 ****P ≤ 0.0001 (Figure 5, Lines 268-270) and (Figure 6, Lines 290-292).

  1. In Figure 7, please combine these two panels and provide a statistical comparison of these two immunogens.

Agree: Combined and the p value added to the legend of the figures. The analysis was statistically significant at the p value 0 ****P ≤ 0.0001 (Lines 316-318).

Round 2

Reviewer 1 Report

Comments and Suggestions for Authors

Thank you for your reply to these questions. Below are some concerns.

1) Line 278: “we recovered 7 mg/ml PUUV rN protein from 1L of E. coli LB culture”.

Please double check the concentration of recovered rN protein, which (7 mg/ml) still seems too high for protein that is prone to aggregation. If it is indeed 7 mg/ml, please provide the total volume, so that we can know the total mass of rN protein recovered from 1L of E. coli culture.

2) There are no error bars in Fig. 7. What does the Positive control stand for in Fig. 7? The data point of positive control (Fig. 7) using N protein as an antigen coated on the ELISA plate is missing.

Author Response

The authors would like to thank the reviewer for the insightful revision of our manuscript. Kindly find point by point the revision of the recommended points. All the suggestions have been incorporated into the main text.

1) Line 278: “we recovered 7 mg/ml PUUV rN protein from 1L of E. coli LB culture”.

Please double check the concentration of recovered rN protein, which (7 mg/ml) still seems too high for protein that is prone to aggregation. If it is indeed 7 mg/ml, please provide the total volume, so that we can know the total mass of rN protein recovered from 1L of E. coli culture.

Agree: After counter checking for the concentration, we found that the 7 mg/ml concentration is correct and the volume of the protein purified from 1 L is 50ul. This concentration was achieved through reconcentration using amicon ultrafiltration cells with a 30 kDa cutoff. The sentence is added to the text Lines 283-285.

2) There are no error bars in Fig. 7. What does the Positive control stand for in Fig. 7? The data point of positive control (Fig. 7) using N protein as an antigen coated on the ELISA plate is missing.

Agree: The positive control is explained Lines 325-326. We did not have a positive control for the immunized mice using N protein as an antigen. The only positive control was in VektoHanta-IgG kit and this control could not be used with N protein as an antigen. The error bars are inserted in fig. 7.

Reviewer 2 Report

Comments and Suggestions for Authors

In this revised manuscript, the authors have made efforts to address most of the concerns I raised in the previous review and have improved the writing in both the results and discussion sections. However, there is still room for improvement in the overall structure and data presentation. Some of concerns are listed below.

Figure 5: The statistical analysis is incorrect.

Figure 6: The data should be combined into a single figure.

Figure 7: Needs to be restructured to accurately reflect the antibody titer dynamics over time for both the first and second immunizations.

Ideally, all these figures should be merged into 2-4 larger figures.

Comments on the Quality of English Language

Acceptible

Author Response

The authors would like to thank the reviewer for the insightful revision of our manuscript. Kindly find point by point the revision of the recommended points. All the suggestions have been incorporated into the main text.

Figure 5: The statistical analysis is incorrect.

Agree: the data was recalculated and the statistics is provided Line 275. The figure was reconstructed.

Figure 6: The data should be combined into a single figure.

Agree: the data was combined and the figure reconstructed.

Figure 7: Needs to be restructured to accurately reflect the antibody titer dynamics over time for both the first and second immunizations.

Ideally, all these figures should be merged into 2-4 larger figures.

Agree: the figures were reconstructed to accurately depict the serum reactivity with N protein and using the kit. We would just like to mention that we did not aim on calculating or measuring the antibody titter or comparing the N protein ELISA (in house) to vektoHanta-IgG kit. We strived to show that the N protein elicited the immune response in mice after immunisation. The serum obtained from immunised mice reacted with both N protein in house ELISA and vektoHanta-IgG kit. So, we could not combine the two graphs but recalculated the data and provided statistics analysis results Line 329.

Round 3

Reviewer 1 Report

Comments and Suggestions for Authors

There is something wrong with the Figure 2.

Author Response

The authors wish to thank the reviewer for the time spent on ur manuscript and his suggestions and recommendations.

Agree: Figure 2 has been updated.